# Evaluating the Impact of Future Seasonal Climate Extremes on Crop Evapotranspiration of Maize in Western Kansas Using a Machine Learning Approach

Kelechi Igwe [1], Vaishali Sharda [1,*] and Trevor Hefley [2]

[1] Carl and Melinda Helwig Department of Biological and Agricultural Engineering, Kansas State University, Manhattan, KS 66502, USA

[2] Department of Statistics, Kansas State University, Manhattan, KS 66502, USA

* Correspondence: vsharda@ksu.edu; Tel.: +1-785-532-2745

**Abstract:** Data-driven technologies are employed in agriculture to optimize the use of limited resources. Crop evapotranspiration (ET) estimates the actual amount of water that crops require at different growth stages, thereby proving to be the essential information needed for precision irrigation. Crop ET is essential in areas like the US High Plains, where farmers rely on groundwater for irrigation. The sustainability of irrigated agriculture in the region is threatened by diminishing groundwater levels, and the increasing frequency of extreme events caused by climate change further exacerbates the situation. These conditions can significantly affect crop ET rates, leading to water stress, which adversely affects crop yields. In this study, we analyze historical climate data using a machine learning model to determine which of the climate extreme indices most influences crop ET. Crop ET is estimated using reference ET derived from the FAO Penman–Monteith equation, which is multiplied with the crop coefficient data estimated from the remotely sensed normalized difference vegetation index (NDVI). We found that the climate extreme indices of consecutive dry days and the mean weekly maximum temperatures most influenced crop ET. It was found that temperature-derived indices influenced crop ET more than precipitation-derived indices. Under the future climate scenarios, we predict that crop ET will increase by 0.4% and 1.7% in the near term, by 3.1% and 5.9% in the middle term, and by 3.8% and 9.6% at the end of the century under low greenhouse gas emission and high greenhouse gas emission scenarios, respectively. These predicted changes in seasonal crop ET can help agricultural producers to make well-informed decisions to optimize groundwater resources.

**Keywords:** extreme weather events; crop evapotranspiration; climate change; machine learning

## 1. Introduction

Crop evapotranspiration (ET), when estimated accurately, can enable agricultural producers to make effective irrigation management decisions. In areas such as the High Plains region, agricultural production relies heavily on groundwater resources [1]. However, due to the declining amount of yearly precipitation, groundwater withdrawals have greatly surpassed recharge, posing a threat to the sustainability of agriculture in the future [2–5]. Since crop ET is the estimate of the actual amount of water required by the crops at any given time during growing season, the effective use of crop ET information can help mitigate this problem. Therefore, when incorporated into irrigation scheduling, crop ET data can help agricultural producers optimize the use of limited groundwater resources and ensure its sustainability by reducing the rate of groundwater pumping while also meeting crop water needs [6,7].

Crop ET rates and amounts, however, are affected by the variability of growing season weather conditions. More specifically, the increased frequency and intensity of the occurrence of extreme climate events caused by climate change influences in-season

variation in the crop water requirement [8]. Climate extremes are instances of harsh weather conditions, such as heat waves and highly variable precipitation. They have become a global concern to agricultural researchers and producers, especially due to the extent of the effect they can have on the hydrological cycle and on crop productivity [9]. Many studies have assessed the impact of climate extremes on the end-of-season yield of crops and discovered that relationships exist between climate extremes and crop yields [10–12]. For example, researchers [13,14] report that extreme heat occurrences, which are conditions where the maximum temperature during the growing season exceeds 30 °C, have a negative effect on maize yield. This is because prolonged exposure to high temperatures induces water stress by increasing the rate of soil evaporation and plant transpiration, as plants are then forced to close their stomata in order to prevent desiccation [15]. A decreased yield at the end of the growing season is therefore likely if sufficient water is not provided to relieve the water stress brought on by these extreme temperature conditions. These extreme weather conditions may even have more devastating effects on crop productivity, particularly if they occur at growth stages where the crops are most vulnerable to water stress [16]. In an experimental study, Hatfield and Prueger [17] reported that the reproductive stage of maize is most sensitive to warmer temperatures, and when compared to maize growth under normal temperatures, these warmer temperatures can reduce the end-of-season yield by as much as 80% to 90%. A similar study [18] also reported that with the exposure of crops to climate extremes, like 24 h under temperatures above 33 °C, the biomass yield is reduced by up to 3%. Maize is highly susceptible to water stress [19], and it needs more adequate watering during its vegetative and tasseling stages in order to prevent significant yield decline at the end of its growing season [20]. However, under well-managed conditions, maize responds quite well to irrigation in terms of production, yielding 673 kg/ha to 1009 kg/ha for every 25 mm of water (one inch) applied. This makes maize the most heavily irrigated crop in Kansas, covering over half of the state's three million acres of maize-producing land. Rogers et al. [21] report that the full-season variety types of maize need between 500 mm and 800 mm of water, often at a rate of 9 mm to 13 mm per day, depending on the weather during the growing season.

Climate-driven variability in in-season weather conditions will likely impact maize's average crop water needs. Rising annual and seasonal temperatures resulting from increased greenhouse gas emissions due to human activities [22] are already being reported in the Midwest [23] and other parts of the world [24]. These observed changes are expected, given that future climate estimates from global climate models (GCMs) show an upward trajectory in global warming and aridity in the future [25]. GCMs are models developed based on the knowledge of the earth's system and how its components interact. Using historically observed data and scenarios that depict the levels of greenhouse gas emissions from human activities, these GCMs generate data for various climate parameters like temperature, precipitation, humidity, etc., at various regional and temporal scales. Four gas emission scenarios, referred to as representative concentration pathways (RCPs), were developed and standardized by the Intergovernmental Panel on Climate Change (IPCC) in the IPCC's Fifth Assessment Report [26]. Two of these scenarios are commonly used in studies, which are the RCP4.5 scenario—a future in which mitigation efforts are implemented to reduce greenhouse gas emissions—and the RCP8.5 scenario, which depicts a future in which greenhouse gas emissions continue to rise all through the twenty-first century. The scenarios serve as input data to over twenty GCMs that have been developed by the Coupled Model Intercomparison Project (CMIP). The predictions from the GCMs are usually compared with historically observed data to estimate the severity of climate change to be anticipated in the future. The changes projected by these GCMs pose a significant threat to the sustainability of agriculture in semi-arid regions like the US High Plains region, particularly since the current limitation in groundwater resources might not be adequate to satisfy a change in the growing season water requirement of maize in the future. Furthermore, Seneviratne et al. [23] reported a likely increase, globally, in the frequency and severity of extreme weather events.

The US High Plains—a portion of the great plains—comprises eight states, which include South Dakota, Nebraska, eastern Colorado, southern Wyoming, western Kansas, northwestern Texas, eastern New Mexico, and northwestern Oklahoma. Together, these states produce more than 50 million tons of grain annually [27]. Agricultural production in the region depends heavily on irrigation activities, which account for approximately 90% of the yearly water use. Irrigation significantly lowers heat stress on maize, which is brought on by extreme climatic conditions during the growing season [28]. Research shows that irrigation increases the overall seasonal biomass yield of crops in the region by 51% [27], which is approximately USD 3 billion today. However, irrigation induces a strain on water resources, as it further depletes the limited groundwater resources that are available [29]. A related study shows that the current groundwater management conditions in the High Plains region are inadequate and may lead to a decrease in maize production by acreage by as much as 60% if no further adaptation strategies are implemented [2]. Furthermore, a long-term trend analysis of climate in western Kansas, which is a part of the High Plains region, revealed that the annual number of frost-free days has increased by 5.2 days [30]. With this rate of warming, future moisture loss from evapotranspiration can rise, which will lead to a corresponding increase in the amount of water needed to irrigate the maize crop. This can eventually impact the crop yield, as the limited amount of groundwater might not be adequate to support the irrigation demand, which is determined by the crop evapotranspiration estimates.

Crop evapotranspiration is often modeled as a function of the reference evapotranspiration $ET_o$, which accounts for most of the environmental influences like temperature, solar radiation, and wind, and the crop coefficients ($K_c$), which account for the effects of crop characteristics like crop height, albedo, canopy resistance, and even evaporation from the soil [31]. $ET_o$ estimation models may be broadly classified into the following: (1) fully physically based models such as the widely accepted Penman–Monteith equation, and the Surface Energy Balance System (SEBS) [32] that incorporates mass and energy conservation principles; (2) semi-physically based models like the Surface Energy Balance Algorithm for Land (SEBAL), the mapping evapotranspiration at high resolution with internalized Calibration (METRIC) model, and the Variable Infiltration Capacity model—often applied in spatiotemporal studies using remote sensing—which combines empirical adjustments with either mass or energy conservation principles [33–35]; and (3) black-box or machine learning models based on artificial neural networks [36,37], adaptive neuro fuzzy computing [38], and genetic algorithms [39], which estimate $ET_o$ by learning patterns and relationships from a given set of input data. Physically based models are computationally demanding as they require a lot of data, and as a result, they can be cost intensive. In contrast, semi-physically based models [33] offer less expensive options, especially when carrying out studies over larger areas.

However, due to recent advances in technologies and processing capabilities, machine learning (ML) models are becoming popular in estimating $ET_o$, and also in studying its associated climatic impacts. This is largely because machine learning models are more effective than other methods at predicting both linear and non-linear relationships, as is often the case for climate-related phenomena [40,41]. ML models such as random forest (RF), support vector machine, artificial neural networks (ANNs), and many more have been applied in predicting crop water demands and crop water stress [42] for various vegetation types [43] and under limited climatic data [44,45]. For example, a comparative study in China [46] evaluated the performance of simple tree-based machine learning models like random forest (RF), extreme gradient boosting (XGBoost), and the M5 model in comparison to other related machine learning models like support vector machines (SVMs). The result indicates that based on determining factors like complexity level, prediction accuracy, stability, and computational costs, the RF model generally provided satisfactory estimates of ETo ($R^2 > 0.9$) in the temperate continental, mountain plateau, and temperate monsoon zones of China. In a similar study [47], the same determining factors were used to compare RF, SVM, and gradient boosting on decision trees with support for categorical

features (Catboost), and it was revealed that in humid regions, the SVM generally exhibited higher levels of accuracy and stability ($R^2 > 0.98$) for ETo estimation under limited climatic data, although it is worth noting that in the study, the RF model also demonstrated a strong performance during the model's training phase, suggesting a potential robustness across different climatic zones. Furthermore, in a comprehensive comparison of eight machine learning models conducted across multiple climatic regions in China [48], the SVM model demonstrated a strong prediction capability. The study categorized the models into different types, including neuron-based models such as generalized regression neural networks (GRNNs), multilayer perceptron neural networks (MLP), and adaptive neuro-fuzzy inference systems (ANFISs); kernel-based models like support vector machines (SVMs) and the kernel-based nonlinear extension of Arps decline model (KNEA); tree-based models like the M5 model tree (M5Tree) and XGBoost, as well as a curve-based model called multivariate adaptive regression spline (MARS). Among these models, the SVM model consistently performed well and exhibited strong predictive capabilities across the evaluated regions in China.

In semi-arid regions like the High Plains region of the United States, ML models have also demonstrated their effectiveness in accurately estimating evapotranspiration. In Kansas, the RF model was observed to effectively capture the temporal and spatial variability of irrigation amounts with a satisfactory accuracy ($R^2 = 0.82$) using hydrometeorological and remote sensing products [49]. Similarly, in Texas, a comparative study between a linear regression model and two more advanced ML models—artificial neural networks (ANNs) and Gaussian process models (GPs)—was conducted to predict daily reference ET. The findings indicate that the GP machine learning model yielded the highest estimation accuracy ($R^2 = 0.95$) [50]. These results highlight the effectiveness of ML models in accurately estimating ET in semi-arid regions, therefore showcasing the potential of capturing the complex relationships associated with ET estimation. However, most of these studies were conducted using mean meteorological variables and only considering the impacts of mean climate change conditions. The seasonal interactions of extreme climate conditions with crop evapotranspiration, which is a crucial component in determining crop yield, have not yet been extensively studied.

To ensure well-informed decision making with regard to irrigation, it is therefore necessary to study the impacts of these climate extremes on crop ET, especially in the face of climate change. There is also a need to quantify how much change in seasonal crop water requirements of a water-intense crop like maize is likely to occur in the future, given the likelihood of an increase in the intensity and duration of extreme climate occurrences. The objectives of this study, therefore, are (i) to analyze historical weather data to determine which climate extremes most influence crop ET, and (ii) to use a machine-learning-based model to quantify the seasonal change in crop ET for future climate change scenarios.

## 2. Data and Methods

### 2.1. Study Area

The southwest portion of Kansas is semi-arid. Its annual minimum and maximum temperatures are averaged at 4 °C and 20 °C, respectively [51]. Also, the mean annual precipitation is less than 500 mm, with a positive trend of only about 2.54 mm per decade [52]. Finney County (latitude = 38.0625° N, longitude = 100.8903° W, elevation of 867 m), which is located in the southwest part of Kansas, was selected as the area of focus for this study. During the growing season—usually from May to October—Finney County (Figure 1) experiences an average precipitation of 349 mm, a maximum temperature of 28.3 °C, and a minimum temperature of 12.3 °C, respectively [53]. Furthermore, only approximately 35% of the reference evapotranspiration is met by the yearly precipitation, and during the growing season, the percentage usually falls between 29 and 48% [53]. The arid condition in the region can be seen in the 30-year time series plot of annual cumulative precipitation and evapotranspiration shown in Figure 1 below. This makes groundwater supplies a crucial resource for agricultural productivity in the study area. There are 1629 wells used

for irrigation; however, the water table in some of the wells has dropped by 15 m since 1950 [54] due to excess pumping.

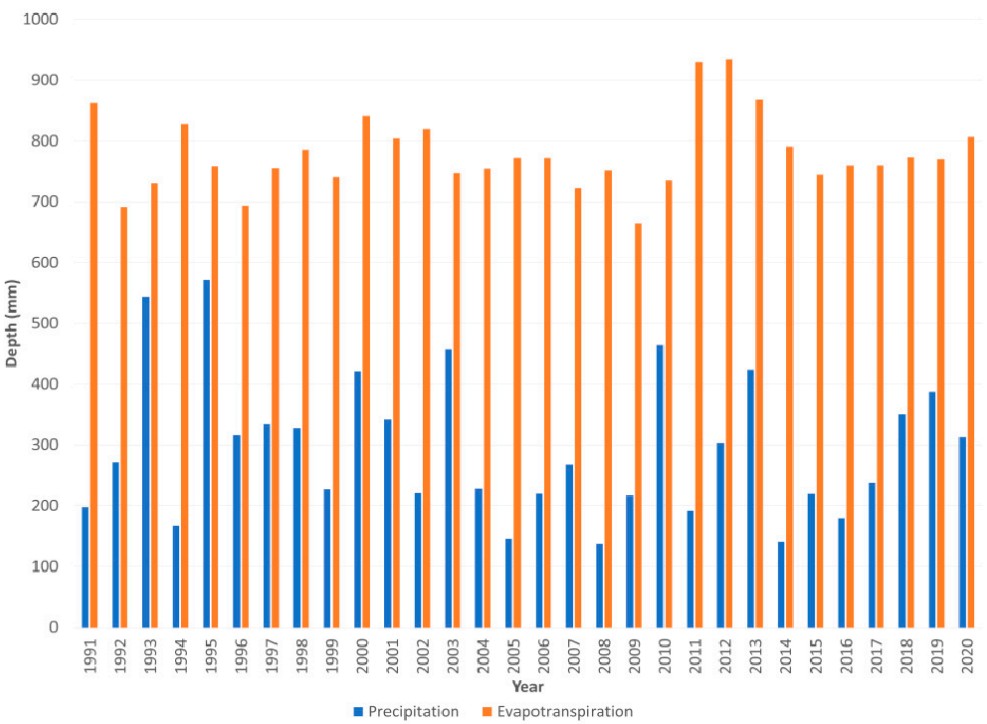

**Figure 1.** A 30-year time series plot of annual cumulative precipitation and evapotranspiration in Finney County, Kansas.

Maize is the second most widely grown crop in Finney County, after winter wheat. According to the 2020 USDA national crop data layer [55], maize covers approximately 50,000 hectares (49,320 hectares) of cultivated land. The three most widely grown crops in Finney County, Kansas, as measured by the area of cultivated land, are shown on the typical county map in Figure 2 below.

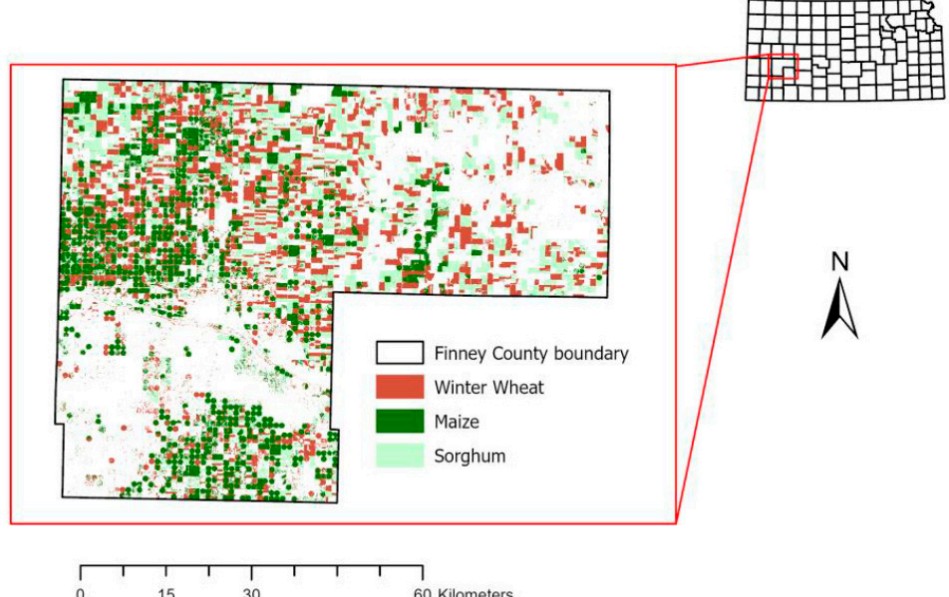

**Figure 2.** Finney County, Kansas showing most cultivated crops by acreage.

### 2.2. Data and Data Sources

#### 2.2.1. Historical Climate Extremes

Weather data comprising daily maximum and minimum temperature and daily precipitation were extracted for Finney County from the High Plains Regional Climate Center (HPRCC) for a 30-year historical time period (from 1991 to 2020). Eleven indices (Table 1) representing agriculturally relevant climate extremes [56] were calculated from HPRCC temperature and precipitation datasets following the procedure recommended by the Expert Team on Climate Change Detection and Indices (ETCCDI) [57,58]. To represent moderate climate extreme indicators, a core set of 27 indices was first created by the ETC-CDI. The initial set of indices developed, however, could not be used in many specialized disciplines, such as agriculture. The indices were then revised by the Expert Team on Sector-Specific Climate Indices (ET-SCI) to make them useful for specific industries like agriculture. Therefore, the indices chosen for this study were those that were considered to be pertinent to agricultural crop production, particularly for growing maize. These indices were also selected based on their capability to be aggregated at a weekly temporal resolution, unlike other indices published by the ETCCDI, which were commonly aggregated at a monthly or annual temporal scale. The calculated indices were aggregated to a weekly temporal scale for the maize growing period to ensure the homogeneity of their temporal resolution with that of the crop ET data. Indices derived from precipitation and temperatures data were categorized as precipitation-based indices and temperature-based indices, respectively. Indices of temperature expressed in percentages—which are commonly called warm spells—were further classified as percentile-based indices. The maize growing season length that was defined in this study for Finney County spanned between May 5th and October 7th as recommended by Araya et al. [53].

**Table 1.** Agriculturally relevant climate extremes adapted from Zhang et al. [58].

| Extreme Indices | Description | Unit |
|---|---|---|
| **Precipitation-based indices** | | |
| Consecutive dry days (CDD) | Maximum length of dry spell: maximum number of consecutive days with RR < 1 mm | day |
| Consecutive wet days (CWD) | Maximum number of consecutive days with precipitation > 1 mm | day |
| Total precipitation (PRPtot) | Weekly total precipitation on wet days (PRPtot) | mm |
| **Temperature-based indices** | | |
| Daily temperature range (DTR) | Daily temperature range; difference between maximum and minimum temperature | °C |
| TM below 10 °C (Tmlt10) | Weekly number of days when TM < 10 °C | day |
| TX of at least 30 °C (TXge30) | Weekly number of days when > 30 °C | day |
| Mean TX (TX_avg) | Mean daily maximum temperature | °C |
| Mean TN (TN_avg) | Mean daily minimum temperature | °C |
| Tropical nights (TR) | Number of days when TN > 20 °C | day |
| **Percentile-based indices** | | |
| Number of hot days (TX90p) | Percentage of days when TX > 90th percentile | % |
| Number of warm nights (TN90p) | Percentage of days when TN > 90th percentile | % |

TN = minimum temperature, TX = maximum temperature, TM = mean temperature.

#### 2.2.2. Estimation of Crop Evapotranspiration

The crop evapotranspiration ($ET_c$) was estimated using two variables, including the reference evapotranspiration ($ET_o$), which was calculated from weather data extracted

from the High Plains Regional Climate Center (HPRCC), and crop coefficient (K$_c$) data (Equation (1)) using the following:

$$ET_c = K_c \times ET_o. \tag{1}$$

where $ET_c$ is the crop evapotranspiration in mmday$^{-1}$; $Kc$ is the crop coefficient; and $ET_o$ is the reference evapotranspiration in mmday$^{-1}$.

This crop coefficient approach in Equation (1) above is recommended by the Food and Agriculture Organization (FAO) for estimating ET$_c$ more accurately from weather datasets when field measurements are not readily available [59]. The step-by-step procedure applied is similar to the one reported by Reyes-González et al. [60], as shown in Figure 3. The ET$_o$ was estimated using the FAO Penman–Monteith equation (Equation (2)) [61] and from weather datasets extracted for Finney County from the HPRCC using the following:

$$ET_o = \frac{0.408\Delta(R_n - G) + \gamma \frac{900}{T+273} u_2 (e_s - e_a)}{\Delta + \gamma(1 + 0.34u_2)}. \tag{2}$$

where $ET_o$ is the reference evapotranspiration in mmday$^{-1}$; $R_n$ is the net radiation at the crop surface in MJm$^{-2}$day$^{-1}$; $T$ is the average daily temperature at 2 m height in °C; $u_2$ is the wind speed at 2 m height in ms$^{-1}$; $(e_s - e_a)$ represent the saturation vapor pressure deficit in kPa; $\Delta$ is the slope vapor pressure curve in kPa °C$^{-1}$; $\gamma$ is the psychometric constant (kPa °C$^{-1}$); and $G$ is the soil heat flux in MJm$^{-2}$day$^{-1}$, which was considered to be equal to zero. This assumption is in accordance with recommendations provided by Allen et al. [61] for calculating $ET_o$ at daily time steps, using a well-watered hypothetical crop of height of 0.12 m, albedo of 0.23, and a fixed surface resistance of 70 sm$^{-1}$, as a reference.

Crop coefficient (K$_c$) of maize crop was estimated from remotely sensed normalized difference vegetation index (NDVI). The FAO's irrigation and drainage paper 56 [31] provide K$_c$ values for maize at the early, midseason, and late season phases of growth. However, these values do not take into consideration the daily and weekly variations of K$_c$ values throughout the course of the full growing season. Several studies [62–65] have reported that a linear relationship exists between the single K$_c$ and the NDVI. Wiederstein et al. [66] compared K$_c$ values of maize, which he estimated from several of these empirical relationships reported by researchers, using NDVI raster images extracted from the Landsat 7 satellite and aerial images captured from uncrewed aircraft. He found that Kamble et al.'s [64] empirical model for K$_c$ produced the best results in Finney County. We therefore applied Kamble et al.'s [64] empirical model in this study to determine K$_c$ values for maize using NDVI data. Surface reflectance images were extracted from both the Landsat 5 satellite for the period of 1991 to 2000 and the Landsat 7 satellite for the years 2001 to 2020. NDVI was computed using the surface reflectance values in near-infrared (NIR) and red regions of the electromagnetic spectrum in the widely applied NDVI index (Equation (3)) [67,68] as follows:

$$NDVI = \frac{NIR - RED}{NIR + RED}. \tag{3}$$

where $NDVI$ is normalized difference vegetation index; $NIR$ is percent reflectance of light in the near-infrared region of the electromagnetic spectrum; $RED$ is percent reflectance of light in the red region of the electromagnetic spectrum.

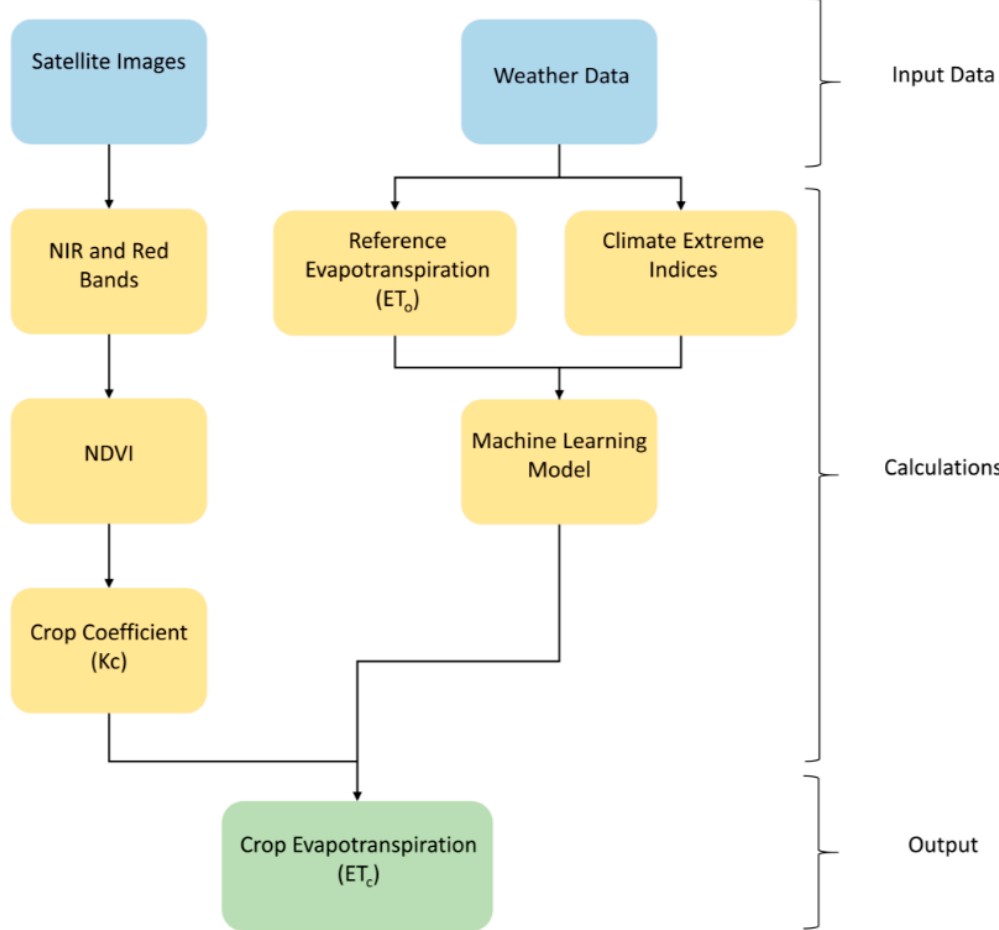

**Figure 3.** Flowchart showing procedure for modeling crop ET from crop coefficient and climate extreme indices. Adapted from Reyes-González et al. [60].

The images obtained from the Landsat satellite sensors are often subject to various influences, including atmospheric effects and bidirectional reflectance distribution function (BRDF), which result from the sun's position, sensor view angle, and the nature of the terrain [69]. Due to the combination of atmospheric influences with these BRDF effects, remote sensing techniques might be particularly challenging, especially in mountainous regions with steep slopes. Although the study area chosen for this research is a relatively flat area with an average slope of less than 2% and an elevation of 867 m above sea level [53], the potential errors that may arise from variations in the elevation and slope, as well as potential atmospheric effects on the quality of the images, were taken into consideration. To address potential errors that may occur due to the BRDF effects, the Landsat satellite surface reflectance products were pre-processed [70] by using approximated BRDF parameters, which are derived from complex algorithms [71]. To further eliminate atmospheric effects, the Landsat surface reflectance images were processed further within the Google Earth Engine (GEE) code editor platform. This process involved filtering out only the days with no cloud cover (i.e., cloud cover = 0%). The filtered images were then clipped to a raster mask of maize-cultivated fields in Finney County using ESRI's ArcGIS pro software. The raster mask containing pixels for maize growing areas was extracted from the cropland data layer (CDL) of the USDA [55]. The clipped images of NDVI were then imported into GEE platform for additional processing. Since the NDVI data were only available for every 16 days because of the Landsat 5 and Landsat 7 thematic mapper satellites' 16-day revisit cycles, further data processing was required in order to provide daily time series for NDVI. Several researchers have adopted statistical smoothing techniques such as the weighted

least-squares linear regression method, Kernel, and Gaussian smoothing techniques to increase the data quality of time series NDVI [72–74]. The Gaussian technique was applied to the NDVI curve using the "gam" function in R [75] in order to extract the NDVI data at daily time steps. The power of polynomial (k = 300) was found to produce the best approximation of the seasonal and temporal pattern of the estimated NDVI data. The smoothened daily NDVI values were then aggregated to weekly time steps to homogenize with the temporal resolution of the climate extreme indices and the evapotranspiration data. The selected NDVI-based $K_c$ model (Equation (4)) [64] was then used to estimate crop coefficient values from the 30-year historical NDVI data.

$$K_c = 1.4571 * NDVI - 0.1725. \qquad (4)$$

The obtained values of Kc were then multiplied with the FAO Penman–Monteith reference ET to obtain the potential crop ET of maize using Equation (1). The estimated crop coefficients were kept constant in historical and future time periods, because the effects of various weather conditions on crop evapotranspiration are largely accounted for by the reference evapotranspiration estimate [31], while the crop-specific characteristics are integrated into the crop coefficient, which is primarily influenced by physiological development stages of the crop, irrigation management, and soil conditions.

### 2.2.3. Future Climate Data and Extreme Indices

Future climate data for Finney County were retrieved from twenty global climate models (GCMs) (Table 2), for a 75-year time period, which were divided into the near-term period (2025–2049), the mid-term period (2050–2074), and the end-of-century period (2075–2099). GCMs are chosen based on how well they perform in capturing the dynamics of various climate phenomena, including precipitation, the pacific oscillations, the dynamics of the El Nino–Southern Oscillation (ENSO), and robust simulations of future climate scenarios. The ability of each GCM to model a particular climate process, however, varies because of the complexity of climate processes [76]. As a result, an ensemble of GCMs is frequently used in climate change research in order to account for the numerous errors and biases that may be present in the individual models. The Coupled Model Intercomparison Project Phase 5 (CMIP5) output of daily statistically downscaled data [77] has been pre-processed by the University of Idaho to extract daily future temperature and precipitation data for the 20 GCMs available at a 1/24-degree spatial resolution. The data were downscaled using the multivariate adaptative constructed analogs (MACA) method [78]. In this study, two representative concentration pathways (RCPs) developed by the intergovernmental panel on climate change were taken into account, including the RCP4.5 scenario, which projects that reduced greenhouse gas emissions will stabilize radiative forcing at about 4.5 Wm$^{-2}$, and the RCP8.5 scenario, which projects that greenhouse gas emissions will increase radiative forcing above 8.5 Wm$^{-2}$ by the end of the century, implying much higher warming in the future under RCP8.5 than under the RCP4.5 scenario. These data were used to calculate the same set of climate extreme indices that were previously calculated using historical weather data. The calculated indices were then used as input data to predict reference evapotranspiration for each GCM and for both the RCP4.5 and RCP8.5 scenarios.

**Table 2.** Statistically downscaled global climate models adapted from relevant studies [78,79].

| S/N | Model Name | Model Agency | Atmosphere Resolution (Lon × Lat) |
|---|---|---|---|
| 1. | bcc-csm1-1_r1i1p1 | Beijing Climate Center, China Meteorological Administration | 2.8 deg × 2.8 deg |
| 2. | CanESM2_r1i1p1 | Canadian Centre for Climate Modeling and Analysis | 2.8 deg × 2.8 deg |
| 3. | CSIRO-Mk3-6-0_r1i1p1 | Commonwealth Scientific and Industrial Research Organization/Queensland Climate Change Centre of Excellence, Australia | 1.8 deg × 1.8 deg |
| 4. | HadGEM2-CC365_r1i1p1 | Met Office Hadley Center, UK | 1.88 deg × 1.25 deg |
| 5. | IPSL-CM5A-LR_r1i1p1 | Institut Pierre Simon Laplace, France | 3.75 deg × 1.8 deg |
| 6. | MIROC-ESM_r1i1p1 | Japan Agency for Marine-Earth Science and Technology, Atmosphere and Ocean Research Institute (University of Tokyo), and National Institute for Environmental Studies | 2.8 deg × 2.8 deg |
| 7. | bcc-csm1-1-m_r1i1p1 | Beijing Climate Center, China Meteorological Administration | 1.12 deg × 1.12 deg |
| 8. | GFDL-ESM2G_r1i1p1 | NOAA Geophysical Fluid Dynamics Laboratory, USA | 2.5 deg × 2.0 deg |
| 9. | HadGEM2-ES365_r1i1p1 | Met Office Hadley Center, UK | 1.88 deg × 1.25 deg |
| 10. | IPSL-CM5A-MR_r1i1p1 | Institut Pierre Simon Laplace, France | 2.5 deg × 1.25 deg |
| 11. | MIROC5_r1i1p1 | Atmosphere and Ocean Research Institute (University of Tokyo), National Institute for Environmental Studies, and Japan Agency for Marine-Earth Science and Technology | 1.4 deg × 1.4 deg |
| 12. | MRI-CGCM3_r1i1p1 | Meteorological Research Institute, Japan | 1.1 deg × 1.1 deg |
| 13. | BNU-ESM_r1i1p1 | College of Global Change and Earth System Science, Beijing Normal University, China | 2.8 deg × 2.8 deg |
| 14. | CNRM-CM5_r1i1p1 | National Centre of Meteorological Research, France | 1.4 deg × 1.4 deg |
| 15. | GFDL-ESM2M_r1i1p1 | NOAA Geophysical Fluid Dynamics Laboratory, USA | 2.5 deg × 2.0 deg |
| 16. | inmcm4_r1i1p1 | Institute for Numerical Mathematics, Russia | 2.0 deg × 1.5 deg |
| 17. | IPSL-CM5B-LR_r1i1p1 | Institut Pierre Simon Laplace, France | 2.75 deg × 1.8 deg |
| 18 | MIROC-ESM-CHEM_r1i1p1 | Japan Agency for Marine-Earth Science and Technology, Atmosphere and Ocean Research Institute (University of Tokyo), and National Institute for Environmental Studies | 2.8 deg × 2.8 deg |
| 19. | NorESM1-M_r1i1p1 | Norwegian Climate Center, Norway | 2.5 deg × 1.9 deg |
| 20. | CCSM4_r6i1p1 | National Center of Atmospheric Research, USA | 1.25 deg × 0.94 deg |

### 2.3. Development of Machine Learning Model and Performance Evaluation

The random forest (RF) regression model [80,81] was used to model the association between climate extremes and crop ET. Studies show that when compared to linear and generalized additive models, the random forest machine learning (ML) model performed better in estimating linear and non-linear relationships involving climate change parameters with minimal errors [41,82]. RFs are regression-type ML techniques that are devised for creating a prediction ensemble utilizing numerous decision trees that are randomly trained on a subset of the input data [80]. Each decision tree is produced by randomly resampling the input data using bootstrapping approach. The same sample of predictors may be chosen for splitting at each node, and the trees run independently of one another [83]. The RF model was validated by splitting the dataset containing the eleven climate extreme indices into two parts, including an initial 70% of the data, which were used to train the model, and the remaining 30% of the data, which were used to test the model. To improve the RF model's performance, the model parameters were adjusted through a process known as "tuning." The parameters tuned for optimum performance were the number of trees (*ntree*), which specifies the number of trees that can be generated, and the *mtry* parameter, which randomly distributes the variables that are used as candidates at each split to form the datasets on which the trees are formed [84]. Using 70% of the data for the parameter tuning resulted in the optimal RF model parameters being the number of variables randomly sampled at each split (*mtry* = 4) and the number of regression trees to build (*ntree* = 500). The coefficient of determination ($R^2$), root mean squared error (RMSE), and mean absolute error (MAE) statistics were used to assess the model's performance accuracy on both the training and the test data. The coefficient of determination given by Equation (5) quantifies the percentage of variance in the response variable that can be explained by the predictor variables [85]. Therefore, it indicates how well the data fit the model. Similarly, when comparing predicted values to actual values in a dataset, the root mean squared error (Equation (6)) reveals the square root of the average squared difference between them. The lower the RMSE, the better the model fits the data [86]. Additionally, the mean absolute error (Equation (7)) is a measure of the average deviation of the predicted values of a model from the actual values [87]. These statistical metrics defined above were utilized to assess the RF model's effectiveness on both the test and training period data.

$$R^2 = 1 - \frac{\sum_{i=1}^{m}(X_i - Y_i)^2}{\sum_{i=1}^{m}(\bar{Y} - Y_i)^2}, \tag{5}$$

$$RMSE = \sqrt{\frac{1}{m}\sum_{i=1}^{m}(X_i - Y_i)^2}, \tag{6}$$

$$MAE = \frac{1}{m}\sum_{i=1}^{m}|X_i - Y_i|. \tag{7}$$

where $X_i$ is the predicted value; $Y_i$ is the actual value; $\bar{Y}$ is the mean of the actual values; $m$ is the number of samples in the equations above.

The influence of each climate extreme index on crop ET was determined using a variable importance plot. The variable importance plot in an RF model illustrates how the elimination of each variable affects the mean squared error (MSE) of the model. Therefore, variables that have a greater influence on the MSE of the model will have a negative impact on evapotranspiration. Additionally, the variable importance plot is very useful for reducing the complexity of machine learning models, since it identifies and selects the most crucial predictors [88]. As a result, it can be used to eliminate variables that have the least impact on the accuracy of the model, further simplifying the model. This process is commonly referred to as feature selection.

*2.4. Future Prediction of Crop Evapotranspiration*

Future prediction of weekly crop $ET_o$ was estimated using the RF model, using climate extreme variables that were extracted from 20 GCMs each, under the two gas emission scenarios considered in the study, and for three future time periods including the near-term (2025–2049), mid-term (2050–2074), and end-of-century (2075–2099) periods. The confidence in the model predictions rely on the performance of the already developed RF model [89], which was assessed using statistical metrics. We computed the average ensemble of the $ET_o$ forecasts from the 20 selected GCMs for each scenario and time period. The ensemble of $ET_o$ forecasts were then multiplied with the crop coefficient values to estimate $ET_c$. An analysis of variance (ANOVA) test [90] was performed at a 95% confidence level to determine if the forecasted mean crop ET values for all the future scenarios were significantly different from the historical time period and from each other. Since an ANOVA test does not provide information about which of the future time periods show significant differences in means from each other and from the historical time period, we further performed Dunnett's test [91] on the forecasted crop ET values for the three time periods using the historical crop ET values as the control group, and then compared each forecasted crop ET values with the crop ET determined from historical weather data so as to quantify the expected percentage change in crop ET in the future.

### 3. Results and Discussion

*3.1. Summary Statistics for Historical Climate Extreme Indices*

Occurrences of unusually extreme weather conditions during the growing season pose a threat to the sustainability of maize production in the High Plains region. This is because they affect the crop evapotranspiration rates, which may have an effect on the region's limited water resources. Climate extreme indices, which represent instances of extreme weather and are thought to affect the evapotranspiration rate of maize, were selected and used as input variables in a random forest model in order to estimate reference evapotranspiration from which to determine the actual crop water requirement. Figure 4 below comprises boxplots of summary statistics showing the weekly variations in the occurrences of these eleven selected climate extreme indices during the maize growing season. Figure 4a shows the boxplot summary of indices derived from the precipitation data, Figure 4b represents the summary of indices derived from the temperature data, and Figure 4c is a summary of the indices of temperature expressed in percentages, represented here as percentile-based indices. The mean weekly maximum number of consecutive dry days (CDDs) over a 30-year period was 4.2 days, with a standard deviation of 2.1 days. Although the CDD showed less variability, as indicated by its coefficient of variation of 48.6%, the upper quartile (75th percentile) of the data revealed that there were instances of up to six consecutive days in the week being completely dry or having a weekly rainfall amount of less than 1 mm. Similarly, the mean weekly total precipitation was 12.8 mm, with an SD of 17.6 mm, while the mean of the consecutive wet days was 1.7 days, with an SD of 1.6 days. The observed significant deviations from their means explain their high coefficient of variations of about 137.7% and 92.8%, respectively. A similar analysis of the distribution of rainfall in western Kansas [92] showed that there is significant variability in the precipitation patterns due to the climate, with an overall tendency of drier conditions. Our analysis also showed high variations in the warm spells, which comprises the weekly percentage of days with a minimum temperature (Tmin) that is greater than the 90th percentile (mean = 26.2%, SD = 29.3%), and the weekly percentage of days with a maximum temperature (Tmax) that is greater than the 90th percentile (mean = 26.2%, SD = 29.3%). Both displayed CVs of 116.3% and 111.9%, respectively. We observed up to 42.2% days in a week where the temperature was greater than the 90th percentile. This is similar to the study by Anandhi et al. [93], which reported that for Kansas, the annual warm spells typically last up to 4 days on average (i.e., up to 60% of days in a week), especially in the summer months. Similarly, other temperature-derived indices such as the mean weekly minimum temperature (mean = 14.2 °C, SD = 4.8 °C), the mean weekly maximum temperature (mean = 29.5 °C, SD =

5.3 °C), the weekly number of days with Tmax greater than 30 °C (mean = 3.7 days, SD = 2.4 days), and the daily temperature range (mean = 15.3 °C, SD = 2.7 °C) in a week all showed less seasonal variation than the precipitation-based indices. Their CVs of 33.6%, 18.1%, 65.7%, 17.8%, and 92.8%, respectively, implied that their individual weekly variations remained relatively constant throughout the entire growing season from year to year. Meanwhile, much higher variations surpassing those of all the other indices were observed in the weekly number of days with a mean temperature of less than 10 °C (mean = 0.1 days, SD = 0.5) and in the tropical nights (mean = 0.6 and SD = 1.1), as indicated by the high coefficient of variations (CV) of 401.7%, and 188.4%, respectively. These extremely high values were expected, however. This is due to the fact that their mean values were very close to zero, causing the CV to become very sensitive to changes in the mean. However, the boxplot summary of the weekly number of days with a mean temperature of less than 10 °C was skewed to the right with an unusually high frequency of outliers, indicating multiple occurrences of cold temperatures with significantly higher magnitudes than the mean observations.

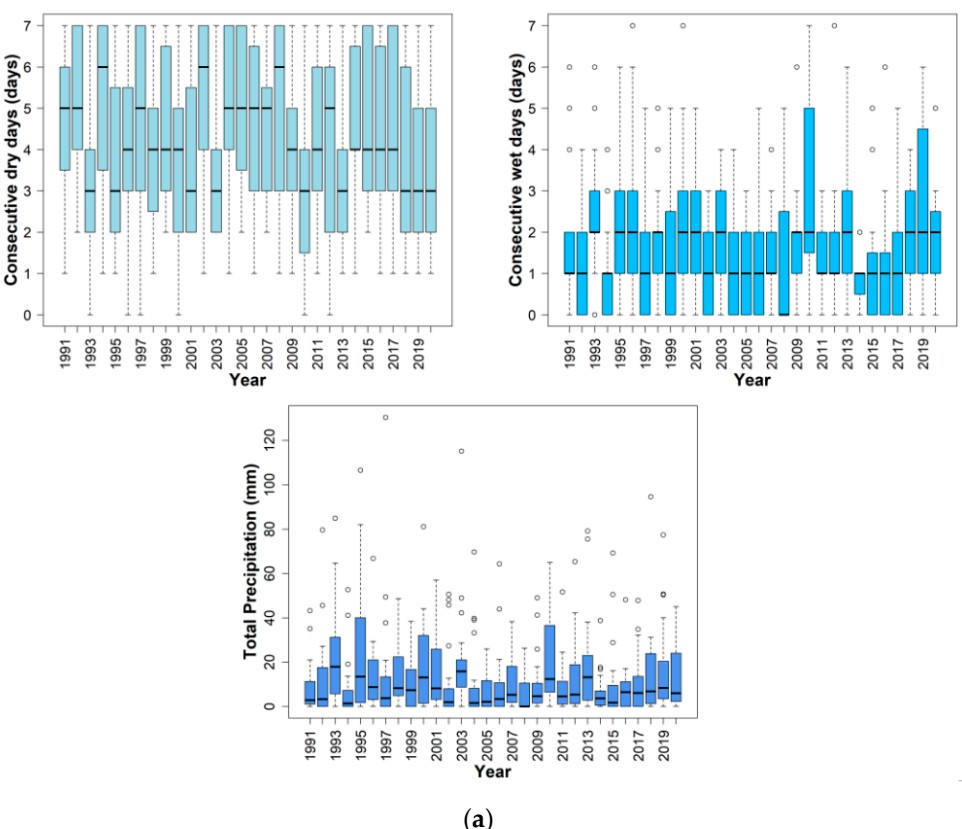

(**a**)

**Figure 4.** *Cont.*

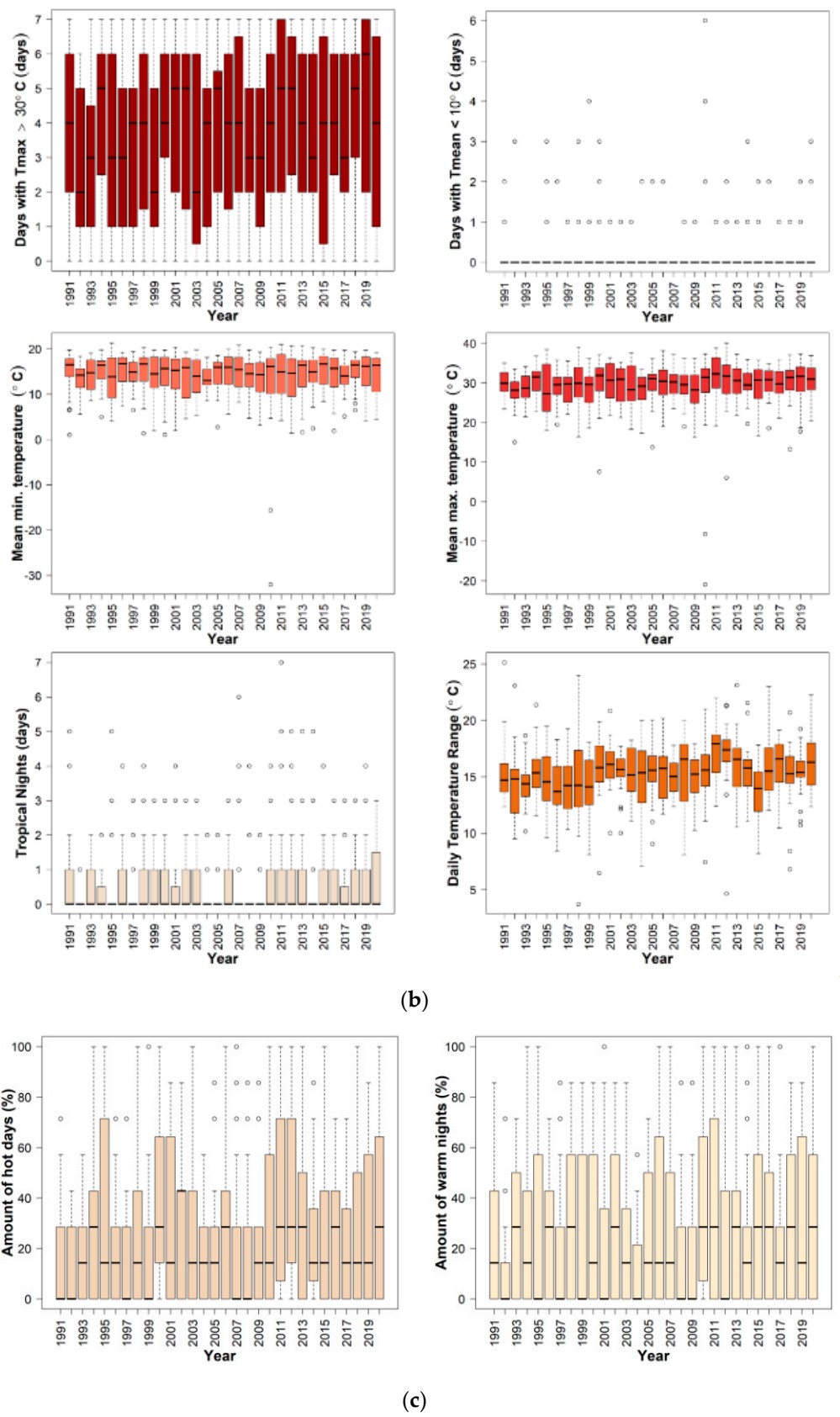

**Figure 4.** Boxplots showing statistical summaries of the selected climate extreme indices: (**a**) precipitation-based indices; (**b**) temperature-based indices; and (**c**) percentile-based indices.

### 3.2. Estimated Crop Coefficient from Maize Pixels

Using Kamble et al.'s [64] linear regression model, the values of the average weekly crop coefficient ($K_c$) for the pixels of maize were determined using Landsat NDVI images. The estimated $K_c$ curve (Figure 5) follows a similar trend with the maize crop coefficient curve reported in the FAO's irrigation and drainage paper 56 [31]. The $K_c$ curve typically starts out with low values at the initial growth stage of maize, and gradually increases over the first 4 weeks (0–30 days) after planting. It reaches its apex at the midseason stage, when the canopy ground cover is between 25% and 60%, before declining to a constant value at the end stage, which typically occurs 12 to 24 weeks (50 to 170 days) after planting. During the early stage of growth, the estimated minimum $K_c$ value was 0.46, while the maximum value observed was 0.65. The observed mean value for the early growth season, which was averaged over the 30-year historical time period, was 0.55, indicating higher values than the $K_c$ value of 0.35, which was reported by the FAO for the initial growth stage. This variation in the values can be attributed to the differing field conditions and management practices of the corn fields in Finney County compared to the conditions in which the FAO $K_c$ values were developed. The FAO provides $K_c$ values based on standard climate conditions involving a sub-humid climate with an average minimum relative humidity of 45% and a moderate windspeed of 2 ms$^{-1}$. However, field conditions often differ significantly from that of the FAO, as seen in a similar study by Singh and Irmak, performed in southcentral Nebraska [94], and also in a study by Abedinpour Meysam in New Delhi, India [95]. Similarly, during the midseason growth stage, when the crop reaches its peak, the values of $K_c$ ranged between 0.55 and 0.99, with peak estimated values of up to 1.06. Meanwhile, the mean peak value of $K_c$ for the 30-year period was 0.90, thus showing a slight under-estimation when compared to the typical FAO value of 1.2. This difference is likely due to the normalization of peak values of $K_c$, resulting from averaging all of the Landsat NDVI values from the pixels of maize-growing areas to produce a single value of NDVI for each day. During the late season, the $K_c$ values ranged between 0.19 and 0.30. The mean value was estimated at 0.24. The FAO value for this phase typically ranges between 0.60 and 0.35 depending on whether the crop was harvested fresh or dried, respectively. These findings suggest that the $K_c$ values for maize are influenced by site-specific variables, and therefore underscores the importance of the site-specific calibration of $K_c$ values for accurate crop ET calculations. For semi-arid areas like Finney County in western Kansas, the FAO irrigation and drainage paper offers adjustment steps, including considering the frequency of growing season irrigation events, to adjust the $K_c$ values appropriately before they are used to calculate the actual crop ET.

### 3.3. RF Model Performance Evaluation

After training the RF ML model using 70% of the entire data and testing it on the remaining 30% of the data, our analysis showed that these selected climate extreme indices explained up to 70% variability ($R^2 = 0.70$) in crop ET on the training data and up to 71% variability ($R^2 = 0.71$) on the test data (Figure 6), implying a satisfactorily robust model. However, we observed that the $R^2$ also increased to about 80% when more variables of mean weather conditions such as the solar radiation and wind speed were added as inputs to the model. It was therefore evident that the $R^2$ is not always a good metric for evaluating the accuracy of the model, even though it has been recommended and widely accepted by researchers as a metric for evaluating machine learning models [96]. The training set's RMSE and MAE values were observed to be 5.73 mm and 4.41 mm, respectively, while for the test data, the RMSE and MAE values were 5.35 mm and 4.30 mm, respectively. A comparison between the observed evapotranspiration from the test data and the predicted evapotranspiration revealed that both very low and large levels of ET were underestimated by the model. This is probably because RF averages values at each node's end when building decision trees, causing extremely high values to be averaged with low values. This was observed in a similar study by Khanal et al. [82].

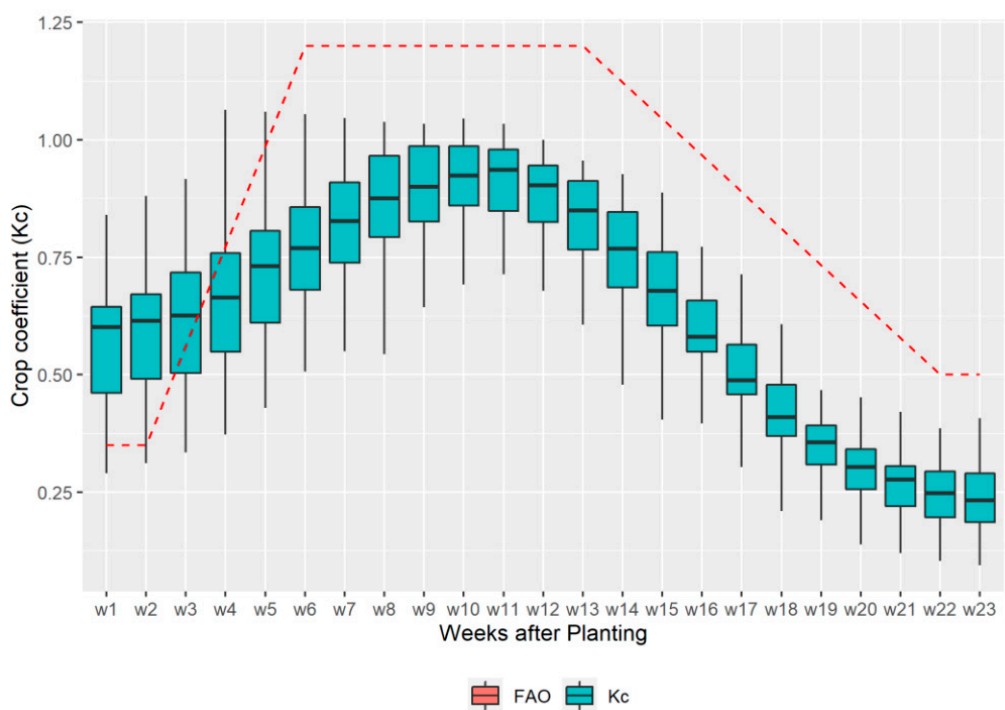

**Figure 5.** Comparison of crop coefficients ($K_c$) estimated from Landsat NDVI using a linear model with the standardized $K_c$ values of FAO.

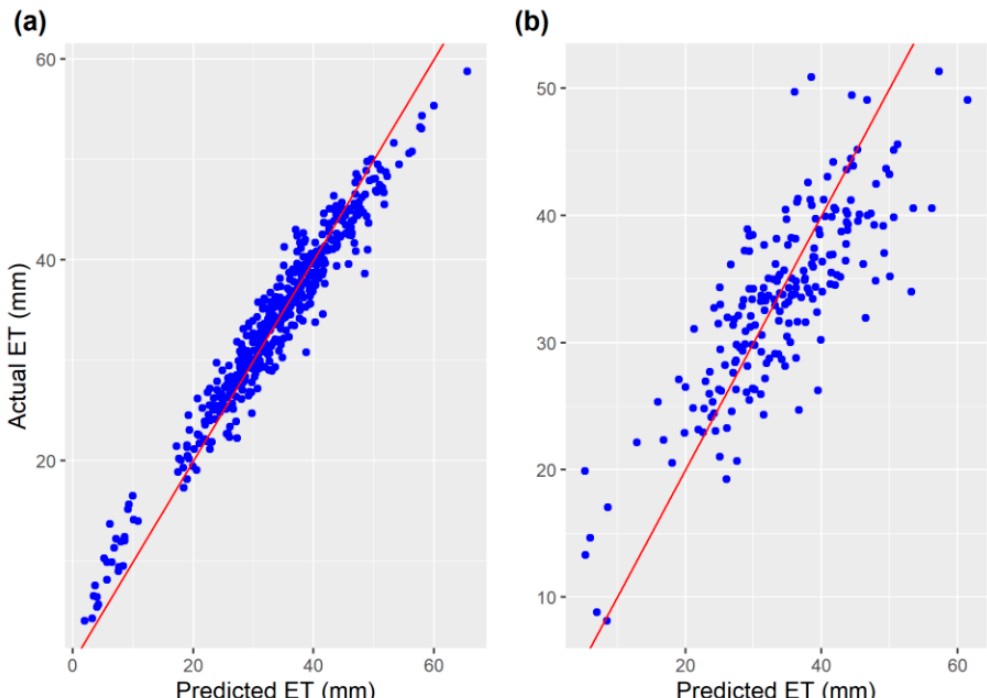

**Figure 6.** Plot of actual evapotranspiration against predicted evapotranspiration: (**a**) training data and (**b**) test data.

### 3.4. Climate Extreme Indices Influencing Crop Evapotranspiration

Figure 7 is a variable importance plot that shows the influence of each climate extreme index on reference evapotranspiration. The indices were ranked based on their individual influences on the mean squared error (MSE) of the random forest (RF) model. We found that by removing the index representing the maximum number of consecutive dry days (CDD), the MSE of the RF model increased by 29.7%, making it the variable with the greatest

influence (Figure 7). This is expected because without rainfall events over a significant number of days, the amount of water loss by evapotranspiration continues to increase. And because there is no recharge of lost moisture in the soil and plants through precipitation, this situation can lead to yield loss, especially if it occurs at growth stages where the maize crop is most sensitive to water stress. Also, we found that the average weekly maximum temperature (tx_avg), when removed from the model, increased the MSE of the model by 27.2%, while the daily temperature range (DTR) increased the model's MSE by 21.3% when eliminated from the model, making the former the second most influential variable, and making the latter the third most influential variable in the model. Climate extreme indices such as the average weekly minimum temperature (tn_avg), the number of hot days (tx90p), the total weekly precipitation (prp_tot), and the weekly number of days with a maximum temperature of more than 30 °C (txge30) were found to influence the MSE of the model by a percentage ranging between 15% and 20%. When these indices were individually removed from the model, the MSE increased by 19.1%, 18.7%, 17.8%, and 17.3%, respectively. Meanwhile, other variables, when removed from the model, only affected the model's MSE by 15% or less; the consecutive wet days (CWD) affected the MSE by 14.2%, the number of warm nights (tn90p) affected the MSE by 10.5%, and the tropical nights (tr) affected the MSE by 7.6%. The number of days with a mean temperature of less than 10 °C (tmlt10) had the least influence on evapotranspiration, increasing the MSE by only 3.5%.

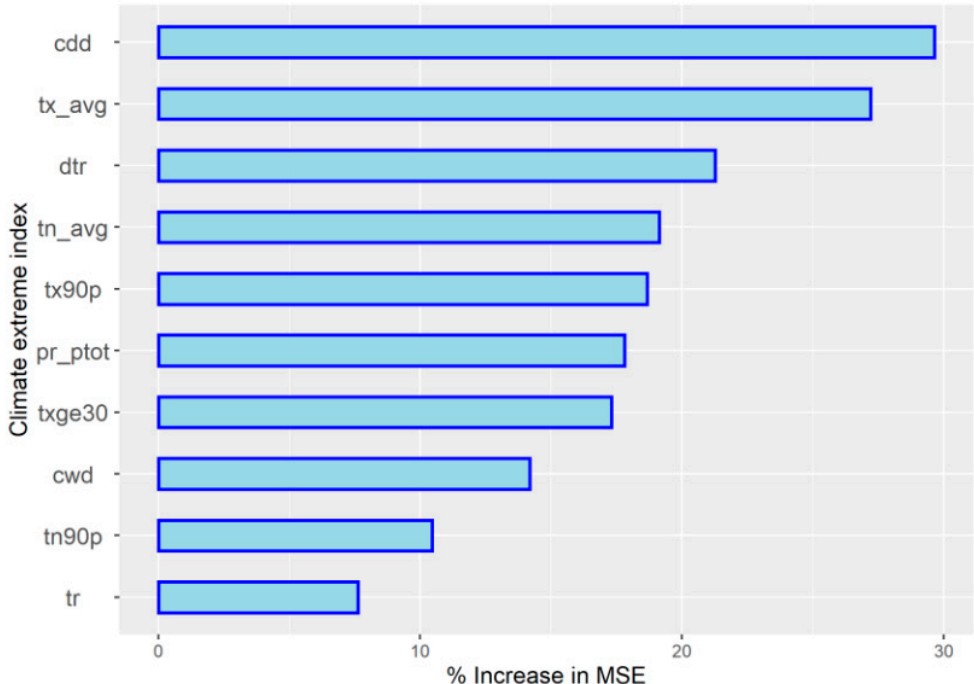

**Figure 7.** Variable importance plot showing influence of each variable on RF model accuracy, where cdd represents the consecutive wet days, tx_avg is the weekly maximum temperature, dtr is the daily temperature range, tn_avg is the average weekly minimum temperature, tx90p is the number of hot days, prp_tot is the total weekly precipitation, txge30 represents the weekly number of days with a maximum temperature of more than 30 °C, cwd represents the consecutive wet days, tn90p represents the number of warm nights, and tr represents the tropical nights.

Although the maximum number of consecutive dry days, which we found to be the single most influential variable, was derived from the precipitation data alone, most of the other derived indices that had the most significant influence on crop ET were observed to be those relating to increasing temperatures. These outcomes are anticipated because temperature continues to be a key driver of ET. This is because more energy becomes

available to turn water into vapor as the temperature rises. Therefore, prolonged periods of high temperature will eventually result in a great loss by evapotranspiration. However, similar studies performed under various climate types in Nigeria [97], Iran, and southwest Asia [98] showed that significant predictors of ET vary depending on the location and field management conditions. In their study, conducted for agricultural lands in midwestern US regions, Talib et al. [99] observed that the key evapotranspiration predictors differed based on whether the fields were rainfed or irrigated. Furthermore, it is also important to note that the variable importance plot also poses some levels of uncertainty as it tends to be biased for a large number of input variables [100], especially when the variables are correlated [101].

### 3.5. Projections of Evapotranspiration in the Future

The ensemble of model predictions of crop ET from 20 global climate models (GCMs) were computed and averaged for both the RCP4.5 and RCP8.5 future climate scenarios, for the near-term (2025–2049), mid-term (2050–2074), and end-of-century (2075–2099) periods. The interquartile ranges of the forecasted weekly crop ET under RCP4.5 (Figure 8) were between 13.4 mm and 31.0 mm in the near term; 13.5 mm and 31.7 mm in the middle term; and 14.3 mm and 31.8 mm towards the end-of-century period. The boxplots for each scenario were slightly skewed to the right, as indicated by the upper whiskers being slightly longer than the lower whiskers. The mean weekly crop ET values were 21.9 mm, 22.6 mm, and 22.9 mm in the near-term, mid-term, and end-of-century periods, respectively.

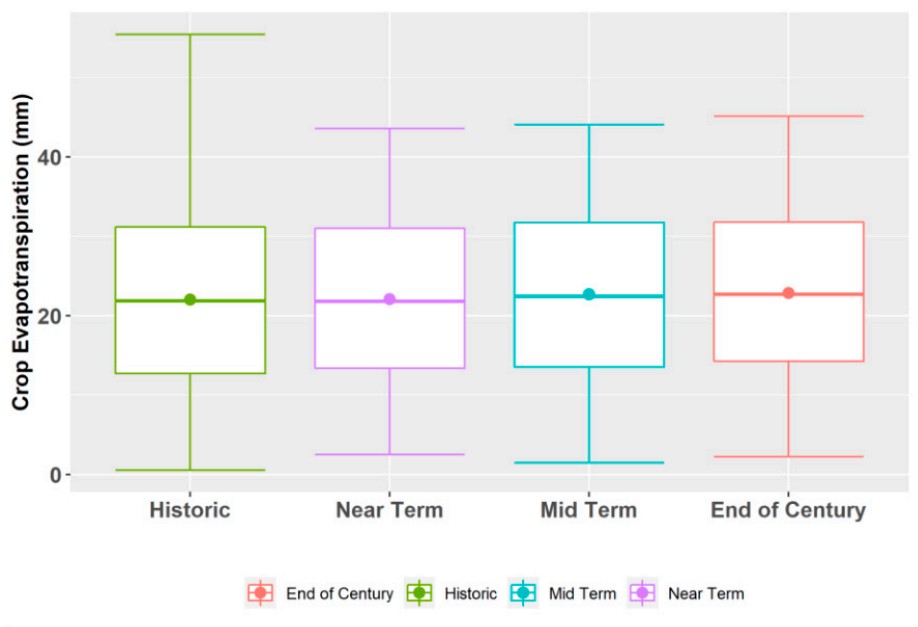

**Figure 8.** Boxplot summary of crop ET predictions under RCP4.5 scenario.

Similarly, under the RCP8.5 scenario (Figure 9), the interquartile ranges of the predicted crop ET values were observed to fall between 13.5 mm and 31.5 mm in the near term; 14.1 mm and 32.3 mm in the middle term; and 15.3 mm and 33.1 mm in the end-of-century period. Although the mean crop ET values gradually increased periodically in the near-term, mid-term, and end-of-century periods, under both the RCP4.5 and RCP8.5 scenarios, the peak estimated value of crop ET ($ET_c$ = 55.4 mm) was higher in the historical time period than the values in each of the future scenarios. This decline in peak values in the future scenario is likely because all the 20 GCMs were ensembled and averaged to obtain a representative value for each scenario. And, as such, all predicted crop ET values were normalized over the individual GCMs. The overall results show an increment in the mean crop ET under both the RCP4.5 and RCP8.5 scenarios, but higher increments in crop ET

were predicted in the mid-term and end-of-century periods than in the near-term period of both the RCP4.5 and RCP8.5 climate scenarios. Also, the wider interquartile ranges of the boxplots during the mid-term and end-of-century periods indicate a higher variability in crop ET values in both periods than in the near term.

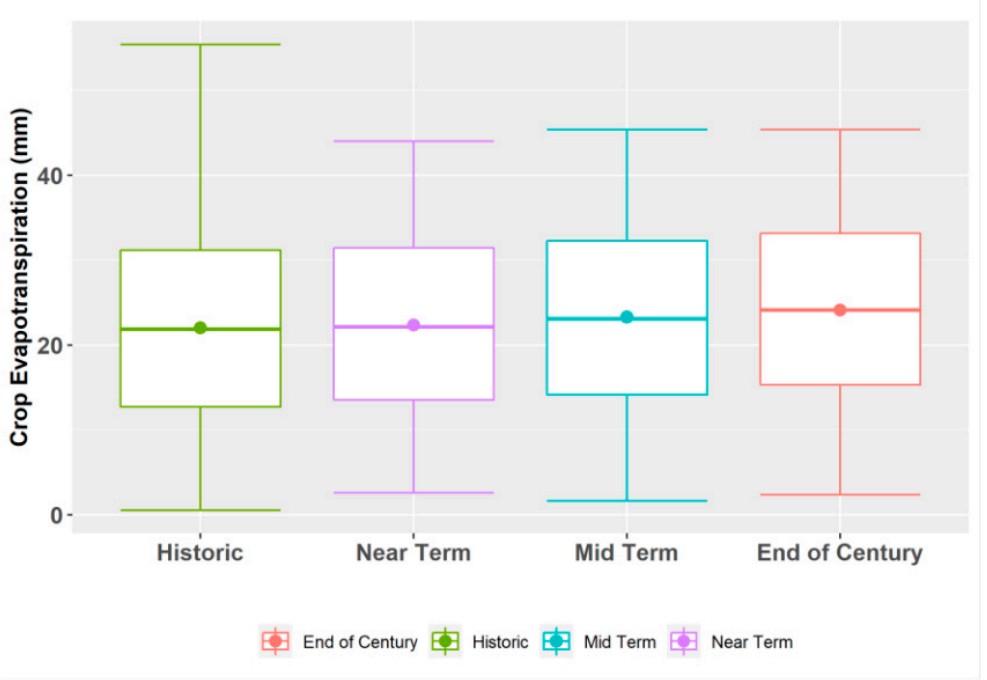

**Figure 9.** Boxplot summary of crop ET predictions under RCP8.5 scenario.

The results of the ANOVA test at a 95% confidence level indicated that there was no statistical difference in the mean values of crop ET ($p$-value > 0.05) under the RCP4.5 scenario. However, an ANOVA test under the RCP8.5 scenario indicated that there were statistically significant differences ($p$-value < 0.05) in the means of the predicted ET values. But since an ANOVA test does not provide information about which of the future time periods under RCP8.5 showed significant differences in the means from each other and from the historical time period, we further performed Dunnett's test on the crop ET values for the three time periods, using the historical crop ET values as the control group. We found that the end-of-century crop ET values were significantly different from the near-term period values, but not significantly different ($p$-value > 0.05) from the mid-term period values. The plot below (Figure 10) shows the weekly crop ET values averaged over a 25-year period for the historical period and the predicted crop ET values for the 75-year future time periods for both the RCP4.5 and RCP8.5 scenarios.

Overall, when compared with the historical data (Figure 11), the results of the ET predictions under RCP4.5 showed a 0.4% increase (mean = 22.1 mm) in the weekly ET in the near term; a 3.1% increase (mean = 22.7 mm) in the middle term; and a 3.8% increase (mean = 22.8 mm) at the end-of-century period. Under RCP8.5, the results of the predicted ET showed a 1.7% increase (mean = 22.4 mm) in the weekly ET in the near term; a 5.9% increase (mean = 23.3 mm) in the middle term; and a 9.6% increase (mean = 24.1 mm) at the end-of-century period. The observed higher increase in crop ET under the RCP8.5 scenario aligns with similar predictions of ET under global temperature rise, caused by high greenhouse gas emissions, leading to greater loss via evapotranspiration [102]. Using an ensemble of three GCMs at three locations in Ethiopia, Gurara et al. [103] projected that by the end of the 21st century, there will be an increase in potential evapotranspiration by an amount between the range of 21.1% and 41% compared to the historical time period under the RCP8.5 scenario.

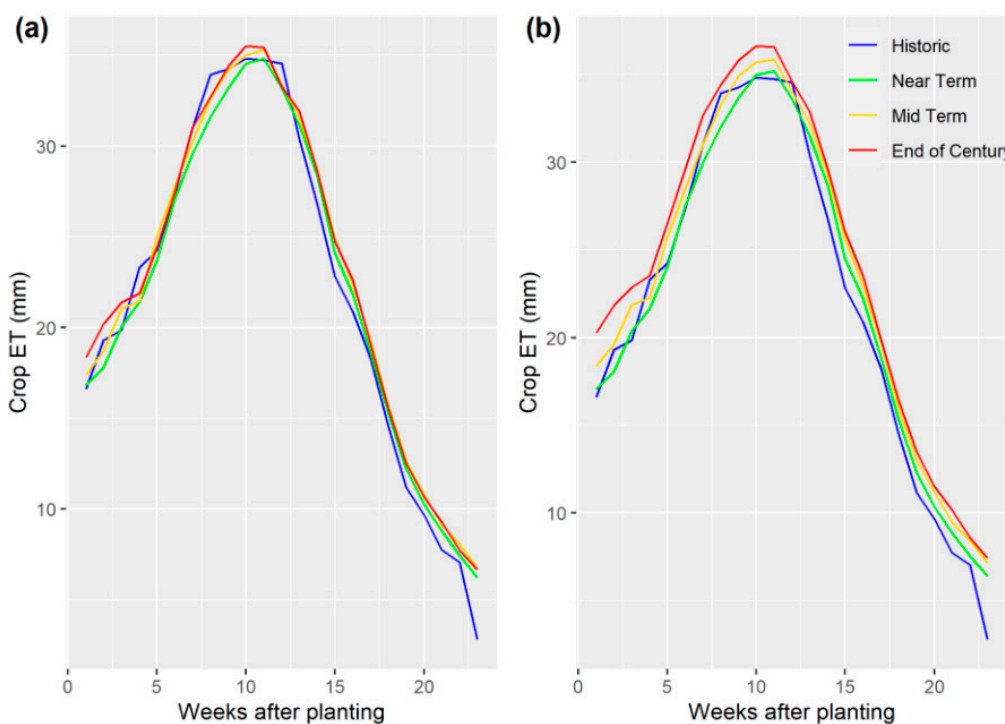

**Figure 10.** Predicted average weekly crop ET values for (**a**) RCP4.5 and (**b**) RCP8.5 scenarios.

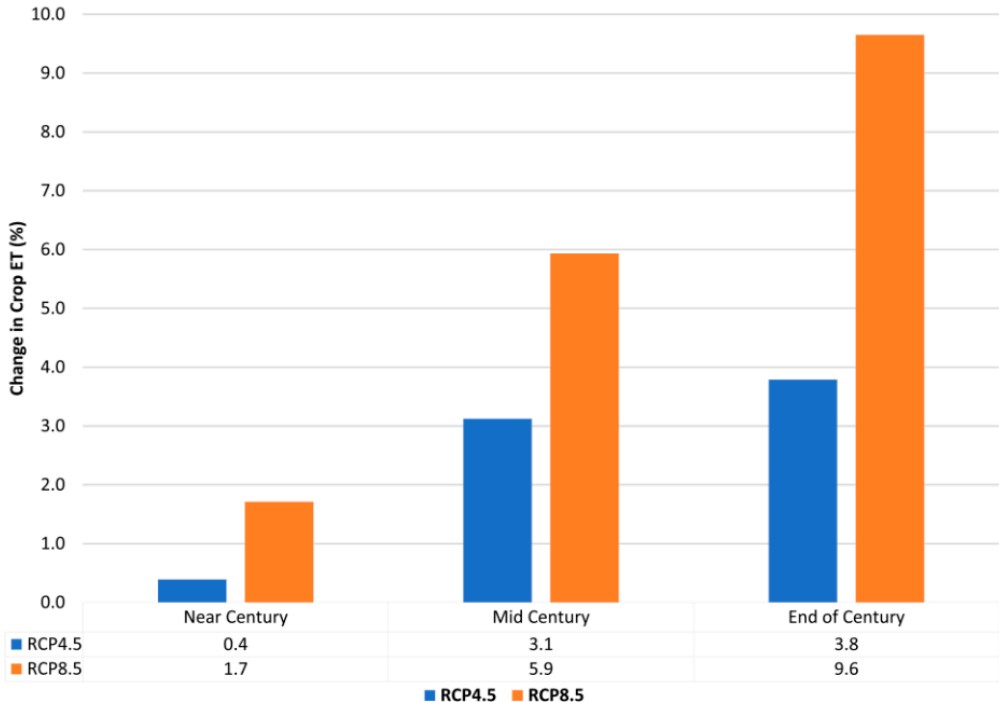

**Figure 11.** Percentage of change in crop evapotranspiration under both RCP4.5 and RCP8.5 scenarios.

## 4. Conclusions and Recommendations

This study analyzes historical data using a random forest model to determine which climate extreme indices most influenced crop evapotranspiration (crop ET) during the growing season. We found that crop ET was most influenced by the maximum number of consecutive dry days and mean weekly maximum temperatures. Our analysis demonstrated that climate extremes can have a substantial effect on agricultural crop water

demand. Given the current limitation of water resources in western Kansas, the persistence of these extreme conditions might lead to an inability to meet the crop water demands of the maize crop. This has critical implications, as inadequate water supply might impact crop productivity, and ultimately, food security. Although crop ET was reported to vary greatly based on a number of other factors such as the planting date and the irrigation management [48], it is advised that robust frameworks be put in place to monitor growing season climate extremes and understand their likely impacts so as to develop adaptation and mitigation strategies against the impacts.

Furthermore, this study provides future projections of crop ET under two representative concentration pathway scenarios, RCP4.5 and RCP8.5. The projections, based on the ensemble of 20 downscaled GCMs, revealed that crop ET would increase significantly under both scenarios in the near-century, mid-term, and end-of-century timeframes. More significant increases were predicted under the RCP8.5 scenario, emphasizing the necessity of reducing greenhouse gas emissions to lessen the effects of climate change on agriculture. In comparison to the historical time period, an average ensemble of the models under RCP4.5 indicated an increase in the weekly ET by 0.4% in the near-term period, 3.1% in the mid-term period, and 3.8% by the end of the century, while predictions under the RCP8.5 scenario indicated an increase in the weekly ET by 1.7% in the near-term period, 5.9% in the mid-term period, and 9.6% by the end of the century. The overall results of this study imply a steady potential increase in crop water requirement in the future. With the aid of these predicted crop ET data and the anticipated changes in crop ET during the growing season, agricultural producers can be better informed in developing strategies to optimize their use of the limited water resources, particularly where limited water rights are allotted to producers. Some of the strategies may include adopting more efficient irrigation techniques, cultivating maize cultivars that are more drought tolerant, and other precision agriculture techniques that could be relevant in increasing crop productivity while reducing groundwater pumping. Additionally, it is imperative to assess the current management practices in the region to evaluate their adaptability to extreme climatic conditions. By identifying areas for improvement and implementing more efficient technologies, farmers and stakeholders can optimize production while reducing the vulnerability to climate extremes and ensuring the long-term sustainability of agriculture.

The effects of these extremes on evapotranspiration, however, were examined by considering the individual influence of each extreme index. However, some of these extreme conditions frequently take place in quick succession or all at once [104]. These situations are frequently referred to as compound extremes. Recent studies [33,105] have reported that the combined effects of dry periods and warming on the evaporative demand are likely responsible for changes in groundwater levels. Therefore, studying these compound extremes might prove to be valuable since their combined effects can have a greater impact on crop ET, and eventually, the end-of-season crop yield. Also, some indices such as the Combined Terrestrial Evapotranspiration Index (CTEI) [106], the Evaporative Demand Drought Index (EDDI), and the Standardized Precipitation Evapotranspiration Index (SPEI) use anomalies in atmospheric evaporative demand and precipitation to establish a relationship with climate extreme conditions, especially droughts. These indices might better estimate the nature of relationships between climate extremes and crop water needs.

**Author Contributions:** Conceptualization, K.I. and V.S.; methodology, K.I. and V.S.; writing—original draft preparation, K.I.; writing—review and editing, K.I., V.S. and T.H.; visualization, K.I.; supervision, V.S. and T.H.; funding acquisition, V.S. All authors have read and agreed to the published version of the manuscript.

**Funding:** This research received no external funding.

**Data Availability Statement:** The data analyzed in this study are available from the authors upon request.

**Acknowledgments:** The authors would like to acknowledge the support of Carl and Melinda Helwig from the Department of Biological and Agricultural Engineering at Kansas State University.

**Conflicts of Interest:** The authors declare no conflict of interest.

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
