# Peer review of "Evaluating the Impact of Future Seasonal Climate Extremes on Crop Evapotranspiration of Maize in Western Kansas Using a Machine Learning Approach"

_land, doi:10.3390/land12081500_

Round 1
Reviewer 1 Report
This is a well written and constructed paper. Crop ET is very important for scheduling irrigation decisions and calculating how much water a crop will need. The paper has used the NDVI, KC to calculate this for maize in the USA mid west. The paper has then looked at some climate change scenarios (4.5 and 8.5) to examine how climate extremes in the future may impact crop growth. Knowing this information now will help plant breeders, agronomists, water managers and farmers develop long term climate change adaptation strategies. In this case it was found temperature derived indices influenced crop ET more than Precipitation.
(I did notice a couple of instances where units probably should be in metric rather than bushels. A couple of the references need a format review)
Author Response
Please find the response attached in the rebuttal document.

Reviewer 2 Report
Comments are attached below

Nil
Author Response
Please find our response attached in the rebuttal document.

Reviewer 3 Report
My main concern with the paper is the below average quality of figures. I am sure this can be improved. Following are few other comments.
Lines 64-65 : a yield decrease of up to 3% -- is this statistically significant, since there would be some error involved in the measurement of crop bio mass yield.
Line 106: yield increase of 51% with irrigation, compared to what? non-irrigated maize, I guess?
Lines 118-132: The models employed in this research, the machine learning models, are not well described in the introduction section, especially when compared to the water management issues related to growing corn in the High Plains of Kansas.
Figure 1: the quality of presentation could be much improved. It is hard to understand the variation of crop ET. Are the ET numbers for the whole season? That is, is it the seasonal crop ET? If it is, then the actual numbers – 1.81 to 2.28 mm? per day?
Figure 3: extremely hard to read and understand.
Most other figures need improvement.
Author Response

(The authors gave the same response as above.)

Round 2
Reviewer 2 Report
I'm writing to provide some comments on the manuscript's revisions. While acknowledging the time and work that went into the study, I would want to draw attention to a few issues with the comments, literature review, and methodology—in particular, the remote sensing-related methodology portion. It seems that the earlier criticisms have not been sufficiently addressed. Despite the fact that the authors have presented their findings and recommendations, there doesn't seem to be much in-depth analysis or discussion of the feedback that has been given. To guarantee a complete comprehension of the research, it would be helpful if the authors could go back and address these concerns in more depth. Second, the article's literature review appears to be insufficient. Establishing a strong foundation by citing pertinent and contemporary literature on the subject at hand is essential. The current evaluation, however, seems to have a narrow focus, leaving out important studies and neglecting to include the most recent study findings. The legitimacy and value of the paper as a whole would be significantly increased by strengthening the literature review. Finally, I'd want to bring attention to the method section, in particular the approach used for satellite image analysis, which still needs to be significantly improved and expanded. I recommend the authors go over the feedback once again and make the necessary changes. I cannot consider it for publishing in its current form without the necessary alterations.
Author Response
Please find our response attached.

Round 3
Reviewer 2 Report
Accept in present form
Minor editing of English language required